# DEMYSTIFYING BLACK-BOX DNN TRAINING PROCESSES THROUGH CONCEPT-MONITOR

## ABSTRACT

Despite the successes of deep neural networks (DNNs) on a broad range of tasks little has been understood of why and how they achieve such victories due to their complex architecture and their opaque black-box training processes. With the goal to unveil the mystery of DNNs, in this work, we propose a general framework called **Concept-Monitor** to uncover the black-box DNN training processes automatically for the first time. Our proposed **Concept-Monitor** enables human-interpretable visualization of the DNN training processes and thus facilitates transparency as well as deeper understanding on how DNNs function and operate along the training iterations. Using **Concept-Monitor**, we are able to observe and compare different training paradigms at ease, including standard training, finetuning, adversarial training and network pruning for Lottery Ticket Hypothesis, which brings new insights on why and how adversarial training and network pruning work and how they modify the network during training. For example, we find that the lottery ticket hypothesis discovers a mask that makes neurons interpretable at initialization, *without* any finetuning, and we also found that adversarially robust models have more neurons relying on color as compared to standard models trained on the same dataset.

## 1   INTRODUCTION

Unprecedented success of deep learning have lead to their rapid applications to a wide range of tasks; however, deep neural networks (DNNs) are also known to be black-box and non-interpretable. To deploy these deep neural network (DNN) models into real-world applications, especially for the safety-critical applications such as healthcare and autonomous driving, it is imperative for us to understand what is going behind the black box. There have been a proliferation of research efforts towards interpretating DNNs and they can be mainly divided into two categories: the first approach focuses on attributing DNN's prediction to the importance of individual-input and identify which pixels or features are important (Zhou et al., 2016; Selvaraju et al., 2019; Sundararajan et al., 2017; Smilkov et al., 2017) while the other approach investigates the functionalities (known as *concept*) of each individual-neuron (Bau et al., 2017a; Mu & Andreas, 2020; Oikarinen & Weng, 2022).

However, most of these methods only focus on examining a DNN model *after* it has been trained, and therefore missing out useful information that could be available in the training process. For example, for a deep learning researcher and engineer, it would be very useful to know:

> *What are the concepts learned by the DNN model and how has the DNN model learnt the concepts along the training process?*

The answer to the above question would be useful in two-fold: (i) it can shed light on why and how DNNs can achieve great success, which could be helpful to inspire new DNN training algorithms; (ii) it can also help to debug DNNs and prevent catastrophic failure if anything goes wrong.

Motivated by the above question, it is the main goal of this work to develop a novel framework **Concept-Monitor**, which makes the black-box DNNs training process become *transparent* and *human-understandable*. Our proposed **Concept-Monitor** is scalable and automated – which are crucial to demystify the opaque DNN training process efficiently and help researchers better understand the training dynamics of the model. More formally, in this paper we provide the following contributions:

- We propose a general framework **Concept-Monitor**, which is the first automatic and efficient pipeline to make the black-box neural network training transparent and interpretable. Our pipeline monitors and tracks the training progress with human-interpretable concepts which provide useful statistics and insights of the DNN model being trained

- We develop a novel universal embedding space which allows us to efficiently track how the neurons' concepts evolve and visualize their semantic evolution through out the training process without the need to re-learn an embedding space proposed in prior work.

- We provide four case studies to analyze various deep learning training paradigms, including training standard deep vision models, the mysterious lottery ticket hypothesis, adversarial robust training and fine-tuning on a medical dataset. With **Concept-Monitor**, we are able to discover new insights into the obscure training process that helps explain some of the empirical observations and hypothesis of the black-box deep learning through the lens of interpretability.

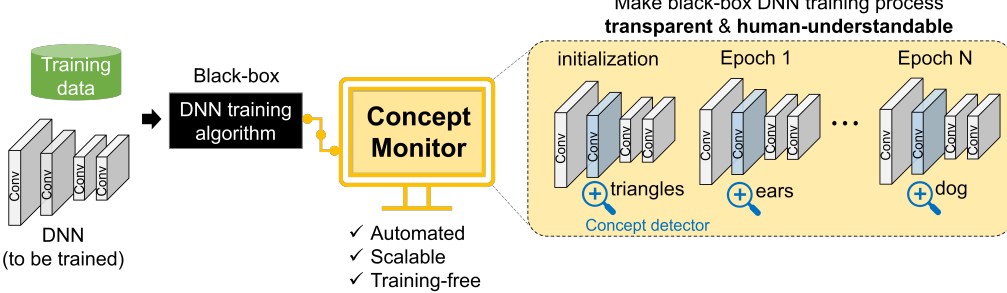

Figure 1: Our proposed **Concept-Monitor** is automated, scalable, training-free and makes DNN training process transparent and human understandable.

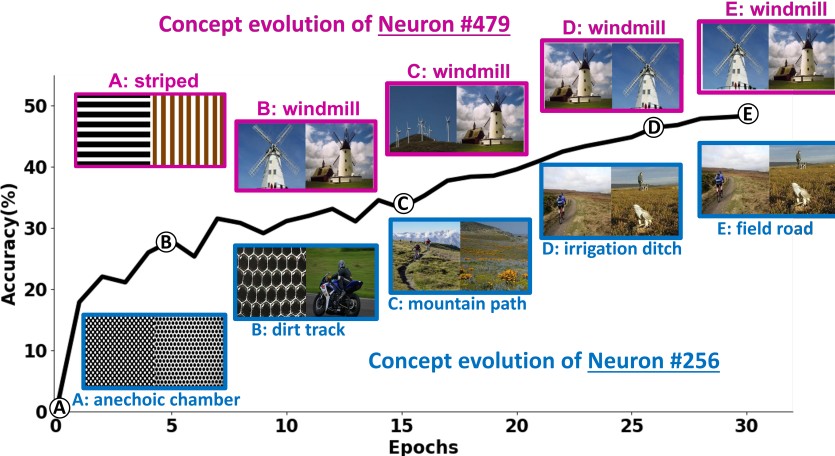

Figure 2: Visualizing the concept evolution of Neuron 256 (blue) and Neuron 479 in Layer 4 (purple) for standard training of Resnet-18 model on Places365 dataset using **Concept-Monitor**

## 2 BACKGROUND AND RELATED WORKS

### 2.1 NEURON-LEVEL INTERPRETABILITY METHODS

Recently, there has been a great interest towards understanding deep neural network models at the *neuron*-level, which is different from mainstream methods that focus on interpreting individual decisions through the input features and pixels (Ribeiro et al., 2016; Lundberg & Lee, 2017; Selvaraju

et al., 2019; Sundararajan et al., 2017). We call this new direction as *neuron*-level interpretability methods and review the representative techniques below. To begin with, the techniques in this direction can be briefly divided into whether it needs to collect a curated annotated concept dataset to dissect DNNs. For the techniques that require a curated probing data labelled with pre-defined concepts, classic methods in this category includes Network dissection and its variation (Bau et al., 2017b; Mu & Andreas, 2020) as well as Test Concept Activation Vector and its variation (Kim et al., 2017; Goyal et al., 2019; Ghorbani et al., 2019). The key idea of Network dissection is to identify *concepts* of neurons by calculating an Intersection over Unit (IoU) score of intermediate activation maps and pre-defined concept masks, while the key idea of Test Concept Activation Vector is to use directional derivatives to quantify the model's sensitivity to the pre-defined concepts.

However, one limitation of this type of approach is the need of a curated probing dataset *annotated* with concept labels which may be expensive and time-consuming to collect. On the other hand, a recent method Clip-Dissect (Oikarinen & Weng, 2022) addresses this challenge by leveraging the paradigm of multi-modal model (Radford et al., 2021) and allows automatic identification of neuron concepts without the need of collecting concept labelled data. We note that these techniques are all compatible to our proposed **Concept-Monitor** to facilitate *automatic* concept monitoring on the DNN training process. In our experiments, we demonstrated the versatility of our **Concept-Monitor** by showing the results with different concept detectors in section 3.2 when we study standard DNN training process.

## 2.2 Understand DNN training dynamics

Most of the existing research has been primarily focused on analyzing models *after* training instead of investigating how the interpretation/concepts change *during* the training DNN process, which is the main focus of our work. We note that there is a recent work Concept-Evo (Park et al., 2022) having the same goal as ours, but their proposed method is very different from our **Concept-Monitor** and their methods have some limitations as discussed below. First, their main idea is to *learn* a universal semantic space for each neuron, using a base model and then project the target model to this space, while we do not need to perform any training. For example, their embedding space uses a base model (VGG19 trained on imagenet) to project target neurons, while we use a pre-trained CLIP (Radford et al., 2021) text encoder to define a universal embedding space. Their methods would be much expensive than ours as they have to redo the learning every time they change the base model or the probing dataset. Second, the approach proposed in Concept-Evo does not associate human-interpretable concepts to the neurons and thus human intervention is required to actually describe each of the neuron, which is another heavy cost (especially when the model size becomes larger and when the training epochs increase) and hard to automate. On the other hand, our method is fully automated and can explicitly provide top $k$ human-understandable concepts for a neuron, which is another advantage of our **Concept-Monitor**.

## 3 Concept-Monitor: A novel, scalable and automated tool to demystify black-box DNN training process

In section 3.1 we detail the key components in **Concept-Monitor** including the concept detector and the universal embedding space. Next in section 3.2, we use **Concept-Monitor** to demystify the standard training process of a deep vision model and discuss the results and insights.

## 3.1 Concept Detector and a Unified Embedding Space

**Concept Detector:** The first part of our method is to use a concept detector ($\phi$) to automatically identify the concept of a neuron at any stage in the training. Given a set of concept words $\mathcal{S}$ and a probing image dataset $\mathcal{D}_{probe}$, a concept detector $\phi$ would return a concept word $w^n$ for a neuron $n$ that maximally activates it. To achieve automatic concept monitoring of a DNN training process, we use two automated neuron-level interpretability tools, Network Dissection (Bau et al., 2017a) and CLIP-Dissect (Oikarinen & Weng, 2022) as the concept detectors in our experiment as a proof-of-concept, and we note that **Concept-Monitor** is compatible with other neuron-level tools as well. Although the technical approach of each concept detector is different, we can actually unify them as a tool calculating a distance metric $d_i^n$ which quantifies neuron $n$'s association with the concept $w_i$.

For example, the distance $d_i^n$ in Network-Dissection (Bau et al., 2017a) is defined to be the IoU score between activation maps and concept masks, while the distance $d_i^n$ in CLIP-dissect (Oikarinen & Weng, 2022) is a measure of the similarity between concept activation matrix and neuron activation maps. Based on this distance metric, we can also define *interpretable neuron*, which are the neurons whose distance to the closest concept word is less than some threshold, i.e. $\min(d_i^n) < \tau$, where the threshold $\tau$ is dependent on the concept detector $\phi$.

**Unified embedding space**: The second part of our method is to define a unified embedding space in order to visually track neurons' evolution. Here we detail the steps to project a neuron $n$ into our unified embedding space.

Step 1: To start with, we use $w_i$ to denote the $i^{th}$ concept in the concept set $\mathcal{S}$ and use $v_i$ to denote the associated text embedding where $v_i = f(w_i)$ with $f$ being the text encoder of a pretrained large language model. We use $\{v_1, v_2, \ldots, v_{|\mathcal{S}|}\}$ as the basis of our semantic space and project neurons on this space using a weighted linear combination of $v_i$ of the neuron's top-$k$ concept words.

Step 2: Let $W_n = [w_{1'}^n, w_{2'}^n \ldots w_{k'}^n]$ be the list of top $k$ concept words for neuron $n$. For each neuron $n$, we can then calculate the embedding $u_n$ using Equation (1) below,

$$u_n = \sum_{i=1}^{k} \lambda_i^n f(w_{i'}^n) \tag{1}$$

where $\lambda_i^n$ is the weight of the concept $w_{i'}$ for describing the neuron $n$ and depends on the concept-detector used. For Network-Dissection, (Bau et al., 2017a), we use the distance vector $d^n = [-IoU_{1'}, -IoU_{2'}, \cdots - IoU_{k'}]$, and for CLIP-Dissect (Oikarinen & Weng, 2022) $d^n = [-h_{1'}, -h_{2'} \cdots - h_{k'}]$ where $h$ is the point-wise mutual information distance metric proposed in the CLIP-dissect paper. We can then calculate $\lambda_i^n$ by fitting a softmax distribution on the corresponding (negative) distance vector have $\lambda_i^n = e^{-d_{i'}} / \sum_{j=1}^{k} e^{-d_{j'}}$. The pseudo code for calculating the unified embedding space is presented in Appendix Algorithm 1

**Remarks:**

1. Note that since our method is general, when using a new concept detector, we only need to change the distance vector $d^n$ associated with that concept detector, which describes how closely related a neuron is to a specific concept.

2. Another benefit of our unified embedding space is that we can project any general concept word $\alpha$ into the same embedding space by calculating its text embedding $f(\alpha)$. This lets us mark the embedding space with concept "anchors" (see the red stars in Fig 3), which are concepts that a researcher thinks would be represented in a well trained model. The researcher can then track whether and which neurons are converging or diverging away from those anchors giving useful feedback during training.

3. Unlike prior work Concept-Evo (Park et al., 2022) which requires training an embedding space every time when a base model changes, our unified semantic space doesn't need to train a base model or learn the image embeddings. Please refer to Table 1 for full comparison between our method and Concept-Evo (Park et al., 2022).

| Method | Training Free | Automatic Concepts | Embedding space Tracking | Embedding space anchors | Flexible probing dataset |
|---|---|---|---|---|---|
| ConceptEvo | No | No | **Yes** | No | No |
| **Concept-Monitor** | **Yes** | **Yes** | **Yes** | **Yes** | **Yes** |

Table 1: Comparison between our method and a recent work Concept-Evo (Park et al., 2022)

## 3.2 CASE STUDY (I) MONITORING STANDARD TRAINING

Now we use **Concept-Monitor** to investigate standard training of ResNet-18 model on Places365 dataset. We investigate the concept evolution of neurons at different epochs in the training using the proposed unified embedding space described in section 3.1.

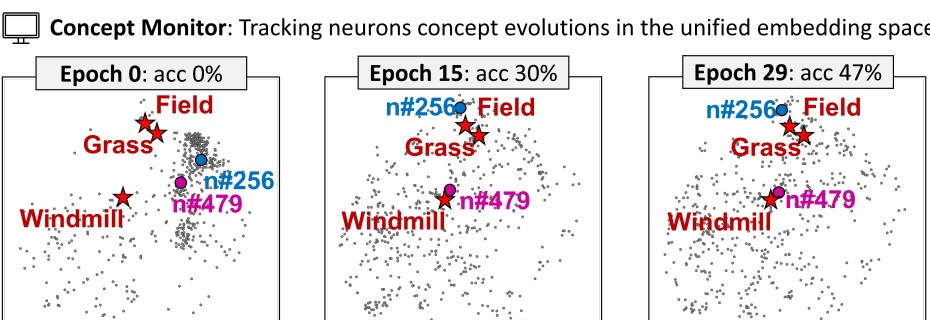

Figure 3: Case study (I): Analysis of Resnet-18 model on Places365 dataset using Broden as the $D_{probe}$. The figure shows our embedding space at three different epochs where each gray dot represents a neuron in layer 4 of the model and red stars represent anchor words. It can be seen that in Epoch 0 neurons are clumped and as the model trains they spread out over the semantic space. Neuron 479 (purple) and neuron 256 (blue) are can be seen to converge to concept anchors "windmill" and "field" consistent with the activation images in Fig 2

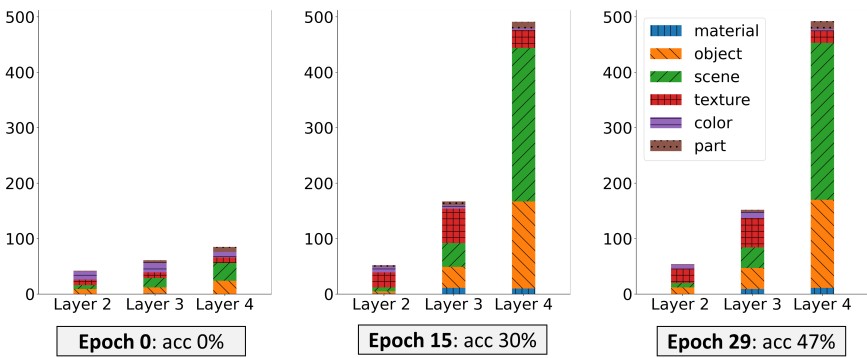

Figure 4: Case study (I) The number of interpretable neurons in each layer separated by the category of concept they encode on y axis vs different layers of the model on x axis plotted for three different epochs. We see that the number of interpretable neurons increase as the model trains and that layer 4 learns more high level features as compared to layer 2 and 3.

**Results and observations.** Our main goal is to inspect the training process and study how the concepts evolve across training and whether there is a correlation between accuracy and concept generalization. The main results are plotted in Figure 3 and we summarize three observations from the standard training below:

1. **Model learns to look at more complex features as training progresses.** As shown in Figure 2, initially neuron 479 is maximally activated by images containing "striped" pattern. As the training progresses, we can see that it starts to learn to identify windmill structures at Epoch 5 and stays the same for the rest of the training. Another examples is neuron 256 which moves from grid pattern like concept of "anechoic chambers" to learning the detect a "field road".

2. **Shallower layers are comparatively more likely to learn low-level features like material and texture while deeper layers learn more nuanced object detectors.** We consider the broad categories of [*Material, Texture, Object, Part, Scene*] to group neurons. These labels were also

used in the original Broden dataset to group the labels. We find that the categories *Scene*, *Object* and *Part* to be concerned with higher level concepts like *Fields* and *Windmills* while *Textures* to be concerned with concepts like *Striped*, *Matted* etc. From Figure 4, its evident that Layer 2 and Layer 3 are learning a lot more low level information than Layer 4.

3. **Concept diversity happens later in the training.** Using the unified embedding space in Figure 3 we can see that the neurons are clumped together in the middle initially (Epoch 0) and as the training progresses they spread out and hence learn more generalized concepts. This suggests that at the initial stage of the training, only a limited number of concepts have been learned and these concepts are similar (close in the embedding space).

**Discussion**: Using our method in standard training, we have seen a correlation between training stage and interpretability of a model. We notice that for a well trained model there is a progression from a low level concepts understanding to higher level conceptual understanding. We propose that an inverse relation might help to improve the model training as well, i.e., a good progression of concepts learnt might indicate a well trained model. Using our methodology, specifically tracking the neuron concept evolution in the unified embedding space, deep learning researchers can meticulously monitor and manage the status of DNN training e.g., they can pause training or modify hyper-parameters when they see neurons grouping up or not spreading out in the semantic space.

**Using another concept detector**: We show that **Concept-Monitor** is able to work with a another concept-detector like Network Dissection (Bau et al., 2017a) by analyzing the same Resnet-18 model trained on Places365 dataset. Our results are in Figure 14 in Appendix C and we can see that the observations are consistent across different concept detectors:shallower layers are more likely to learn low-level features like texture and that model learns more complex features as the training progresses. We also see the embedding space starting from a clump in the center for Epoch 0 and then spreading out indicating the generalization of the concepts learnt.

## 4 CASE STUDIES OF OTHER TRAINING PARADIGMS

In this section, we show the **Concept-Monitor** is versatile and can be used to study various training paradigms to gain insights into how and why they work. We also provide useful observations and insights that could help future researchers better understand these training procedures.

### 4.1 CASE STUDY (II) LOTTERY TICKET HYPOTHESIS

Lottery Ticket Hypothesis (LTH) (Frankle & Carbin, 2018) is a popular method to prune deep neural networks without sacrificing their performance. In this case study, we use **Concept-Monitor** to demystify the success behind LTH in a human understandable way. The main idea of LTH is to use iterative magnitude pruning (IMP) to prune the model iteratively by repeating the steps of training, pruning and rewinding to an initial epoch. LTH hypothesizes the existence of "winning tickets" at initialization which are sub-networks within the network that can be trained to performance equivalent to the original model. However, it was observed that rewinding to initial weight leads to a performance drop and it is better to rewind to an earlier training epoch instead of fully reversing to the initial weights. (Frankle et al., 2019a) attribute this phenomenon to SGD noise in initial training and we will use **Concept-Monitor** to investigate LTH through the lens of interpretability. We train a ResNet18 on CIFAR 10 dataset using IMP in 8 stages. For full details on our experimental setup, please refer to Appendix section A.

We study LTH with three different rewinding stages of IMP: rewinding to initial weights(epoch 0), epoch 5 and epoch 16. We found rewinding to epoch 5 performs better than rewinding to epoch 0 and epoch 16 when the sparsity level is high, which we attribute to be the two extremes of initialization i.e, initialization to epoch 0 in which the model is too noisy or initialization to epoch 16 in which the model has learnt a rigid structure which would need to be rewired by pruning. For instance, when pruned to 2.8% of initial weights, rewind to epoch 5 has 93.78% accuracy as compared to 91.8%, 93.4% of rewinding to epoch 0 and epoch 16 respectively. Rewinding to 0 is inefficient as noted by (Frankle et al., 2019b) and rewinding to epoch 16 doesn't give the model much freedom to adjust to the sparse weights. We use **Concept-Monitor** to track the training process of these 3 different rewinding strategies and plot the results in Fig 5 and Fig 9 in the appendix.

**Observations and Results**:

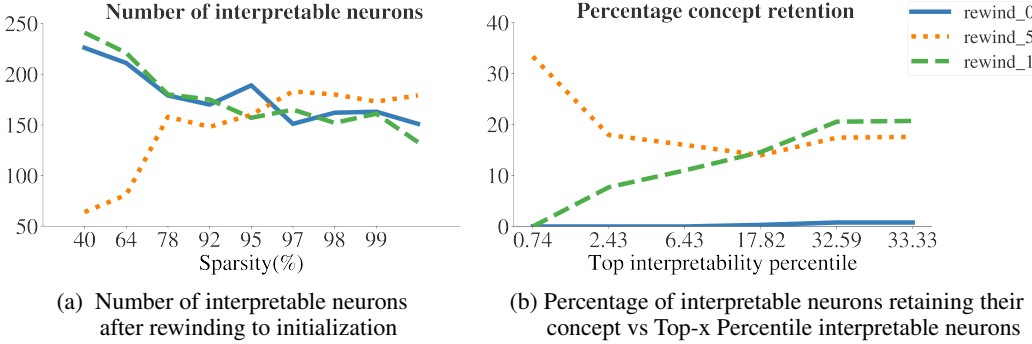

(a) Number of interpretable neurons after rewinding to initialization

(b) Percentage of interpretable neurons retaining their concept vs Top-x Percentile interpretable neurons

Figure 5: Interpretability statistics of our iteratively pruned ResNet-18.

In our analysis, we make the following observations:

1. **Pruning the network learns to encode some concepts without any fine tuning.** Figure 5a shows the number of interpretable neurons in layer 4 of the model *after* rewinding to initialization. We notice the trend that for rewinding to epoch 16 and epoch 5, the number of interpretable neurons decreases as we increase the sparsity, but for rewinding to the initial weights (epoch 0) the number of interpretable neurons *increase*. Since the weights are randomly initialized, the only way there can be a gain in interpretable neurons is through the changes that happen during pruning, i.e. the zeroing out of certain weights in the network. Hence, we believe that there is a possibility that the training is learning to remove connections that are harming the network and this leads to the resultant network to be different than the original (with the only change being that some weights are zero). This leads to some neurons being activated to certain low level concepts and hence our observation of increased interpretable neurons. We note that this phenomenon was also observed by another work (Zhou et al., 2019) which says that IMP zeros out weights that would ultimately go towards zero anyway after training. Hence, they hypothesize that a pruned initial network encodes a portion of the training process itself, which they refer to as "masking is learning". This also explains why we see interpretable neurons with just pruned initial weights.

2. **The percentage of concepts retained through pruning is highest with Epoch 5 rewinding.** Figure 5b plots the percentage of top-x percentile interpretable neurons that retain their concepts throughout the pruning process (y-axis) vs x percentile (x axis). In other words it plots the relation of the interpretability of a neuron to its concept retention.Note that, interpretablity of neurons is dependent on the threshold defined by the concept detector (see section 3.1). We see that for rewind to epoch 16 as we decrease the interpretability the percentage of neurons retaining concepts increases, or the more interpretable neurons are likely to lose their concepts during IMP, while the less interpretable neurons keep their concepts. For rewind to epoch 5 we see that the more interpretable neurons keep their concepts and the retention decreases as the neurons become less interpretable. This leads us to the hypothesis that rewinding to epoch 5 learns concepts that are more general and hence are able to be retained, while rewind to epoch 16 learns concepts that are rigid and the model has to relearn those concepts to preserve accuracy during pruning. This effect is also shown in the accuracy of the models in which rewinding to epoch 5 performs better than rewinding epoch 16 at higher pruning.

**Discussion**: From observation 1, we find that it is very likely that the lottery ticket sparse pruning mask actually encodes learning, which was also suggested in (Zhou et al., 2019) as "masking is learning". It is also noted from observation 2 that certain rewinding points are more suitable to retain concepts, e.g. epoch 5 in our case, and there is a correlation between this and the model performance as noted by the accuracy at higher sparsity.

## 4.2 CASE STUDY (III) ADVERSARIAL TRAINING

DNNs are known to be vulnerable against small perturbations in their inputs (Szegedy et al., 2013). This is problematic as networks can fail unexpectedly after small random or adversarial perturbations which raises concerns over their safety. Fortunately, methods have been developed to defend against

adversarial attacks, most popular of these being Adversarial Training (Madry et al., 2018). This successfully makes networks more robust against such attacks, but comes at a cost of degraded performance on clean test data. In this study, we apply **Concept-Monitor** to adversarial training to better understand how adversarial training changes a network and why standard accuracy suffers. We analyse a ResNet18 model trained on CIFAR10 with and without adversarial training. For full details on our experimental setup please refer to Appendix section A.

**Observations and Results**: Using **Concept-Monitor** we have the following three observations.

1. **Adversarially robust network has less interpretable neurons in late layers, but more in earlier layers.** In Fig 6, we plot the number of interpretable neurons in layer 2-4 at three different training stages. It can be seen at the end of training that 293 out of 512 of the layer 4 neurons are interpretable for standard training while only 215 out of 512 are interpretable for the robustly trained model. For layer 3 it is 91 out of 256 for standard model and 125 out of 256 for the robust model. We observe similar trend for layer 2 neurons, please refer to Fig 16 in Appendix.
2. **Adversarially robust network relies more on colors, less on materials and textures.** When combining concepts detected across 3 layers, we observe that the robust model has a lot more "color" neurons than the standard model (74 vs 15) Figure 16. In contrast, the standard model has 154 neurons detecting "textures" while robust model has only 97, and standard model has 10 "material" neurons compared to only 2 of the robust model. This finding is sensible as detecting textures and materials often relies on high frequency patterns that are easily affected by $l_\infty$ noise therefore the adversarial training forces the model to rely less on them and more on more resilient features like color.
3. **Standard training learns neurons detecting target in the second to last layer while robust training does not.** As seen in Figure 7, the standard network has many neurons detecting its target classes present in the second to last layer. For example, the standard network has 17 interpretable neurons detecting cars and 13 neurons detecting horses in layer4, while the robust network has no layer4 neurons detecting either car or horse.

**Discussion**: We find that adversarial training harms the ability of the network to detect certain concepts that rely on high frequency patterns like texture. Since these patterns are useful for many tasks, losing them may be a significant cause for the degradation in standard performance as observed in the experiments. Another cause for poorer performance of the robust network may be the lack of neurons detecting target class objects in second to last layer, but why this happens is still unclear to us. We believe addressing these two issues may be the key to improving clean accuracy of robust models. On the other hand, the robust network seems to learn more interpretable lower level features perhaps learning a more diverse representation similar to the findings of (Salman et al., 2020) who showed that adversarially robust models have better features for transfer learning.

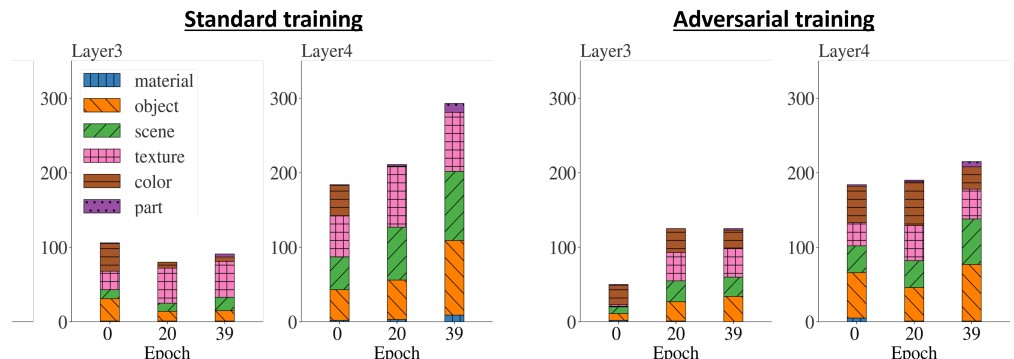

Figure 6: Comparison of the types of concepts learned by standard training compared to PGD-training with $\epsilon$=8/255. We can see the robustly trained model has less interpretable neurons, especially on later layers. Interestingly, there is also a big difference between the types of concepts learned by different models where the PGD trained model still has neurons detecting colors in later layers, while missing concepts detecting textures and materials. Note these figures are the network at the end of each epoch, so epoch 0 is after one epoch of training, not initialization.

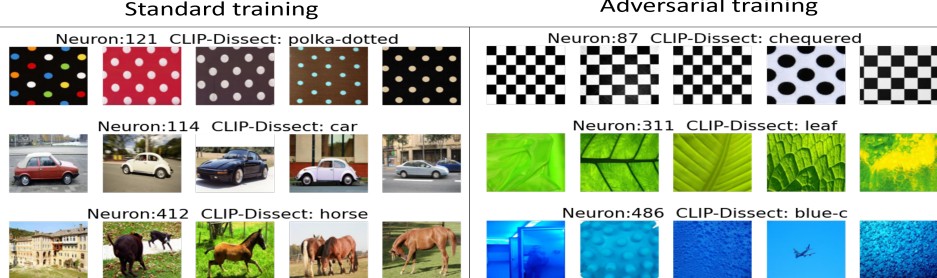

Figure 7: Example neurons detecting the common concepts in layer4 of ResNet-18 for both standard and adversarial training. We can see a large difference between the types of concepts encoded, where standard training learns many neurons already detecting CIFAR-10 classes such as horse or car, most neurons in the robust model are detecting simple patterns like colors.

### 4.3 CASE STUDY (IV) FINE-TUNING ON A MEDICAL DATASET

In this section, we use **Concept-Monitor** to observe the fine-tuning of a pretrained DNN on a diabetric retinopathy dataset (APTOS, 2019). This experiment allows us to test our method on a dataset from a different domain, as well as gather insights on the process of finetuning a pretrained model. The setup details are in Appendix A.

**Observations and results**: We probed the model training at a few intermediate steps. We observe that for the initial weights, as the neurons are pretrained on Imagenet, they show a lot of diverse and high level concepts(as shown in Figure 15 in Appendix). However, as the training progresses we notice that more neurons are getting activated by textural concepts like dots and patterns rather than objects. This is what we expect because as the model gets better at classifying retinopathy images shown in Figure

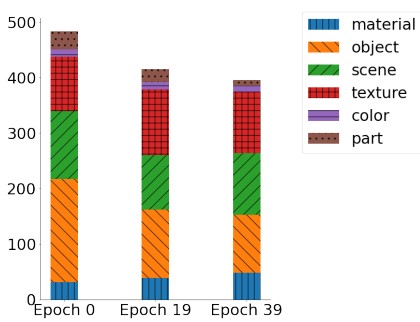

Figure 8: Number of interpretable neurons vs epochs

11, we expect it to rely more on textures and presence of "dots" which is consistent to what we observe here as shown by the top interpretable neurons in epochs 20 and 40 in Figure 15. From Figure 8 we see that the number of interpretable neurons in "object" category decreases as the training progresses while the number of interpretable neurons in the "material" category increases which further confirms our theory that the model learns to focus more on lower level features like material and textures as compared to objects.

## 5 CONCLUSIONS

We have presented **Concept-Monitor**, a novel method to automatically track and monitor neural network training process in a transparent and human-understandable way. With the 4 comprehensive case studies on various deep learning training paradigms, we show that **Concept-Monitor** allows us to better understand the underlying mechanism of standard DNN training, the two alternative training methods, Lottery Ticket Hypothesis and adversarial training, as well as the fine-tuning on medical task. With **Concept-Monitor** we discover that surprisingly lottery ticket hypothesis prunes the network in a way that the neurons are interpretable even at initialization, discovering interpretability hidden in random initialization. Furthermore, we discover that adversarial training causes the hidden neurons to detect more simple concepts like colors while losing representations of materials and target class objects. We also test our method on medical dataset and find that the model learns to focus more on low level features which reflect the medical dataset.

**Reproducibility statement:** We acknowledge the importance of replicating our experiments and for that reason we have explicitly mentioned the implementation details of all our experiments in Appendix A.

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

# APPENDIX

## A. EXPERIMENTAL SETUP

**Standard training (section 3.2)**:

Setup: We train a Resnet-18 model on Places-365 dataset, which contains a lot of diverse classes allowing the DNN model to learn diverse concepts. To reduce the training time, we randomly selected 1000 images for each of the 365 classes and trained for 30 epochs reaching top-1 accuracy of 48.3%. We use batch size of 256 and an initial learning rate of 0.1 with cosine annealing scheduler.

Probing methodology: We use Broden (Bau et al., 2017a) dataset as $\mathcal{D}_{probe}$ and use associated concept labels as a decoupled concept set $\mathcal{S}$. Our embedding space, as described in section 3.1, is computed using CLIP's text embeddings of Broden labels as a basis. For visualizing in a 2-dimension plot, we follow (Park et al., 2022) and use UMAP dimensionality reduction (McInnes et al., 2018), as it preserves inter-point distance in the lower dimensions. We set $k = 5$ in Eq(1), i.e. we use top-5 concepts to compute the embedding.

**Lottery ticket hypothesis experiments (section 4.1)**:

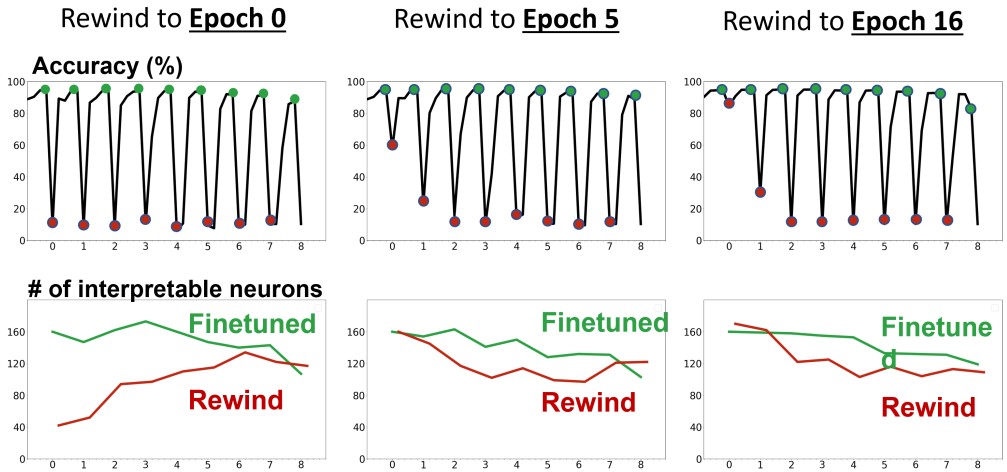

Figure 9: Analysing Layer 4 for Resnet-18 using IMP at different stages of pruning. We observed IMP with rewinding to initial weights, epoch 5 and epoch 16. The top plot is the accuracy vs training stages, The red dots represent accuracy after rewinding, the green dots represent accuracy after finetuning and pruning. The bottom plot is the number of interpretable neurons found at different training stages.

Setup: We train ResNet 18 on CIFAR 10 dataset using IMP as in the LTH paper (Frankle & Carbin, 2018), rewinding to different initial weights. For each stage of IMP we train the model for 160 epochs, prune 40% of the weights and rewind to initialization. We consider rewinding to three different stages: initial weights, epoch 5 and epoch 16, using (Chen et al., 2022) implementation as reference.

Probing methodology: For our $\mathcal{D}_{probe}$, we use CIFAR 100 training dataset and for concept set $\mathcal{S}$ we use broden labels.

**Adversarial Learning experiments (section 4.2)**:

Setup: We perform adversarial training with PGD attacks on a ResNet-18 architecture. We follow reop (Wong et al., 2020) and train the network with $\epsilon = 8/255$ and $l_\infty$ perturbations for 40 epochs. We compare it against a CIFAR-10 network trained using the same exact training setup but no adversarial training. The standard model reaches a final accuracy 94.29%, while the robust model reaches 83.42% accuracy on clean data and 50.00% robust accuracy against a PGD adversarial attack as shown in Figure 10. The standard model expectedly performs really badly on adversarial images.

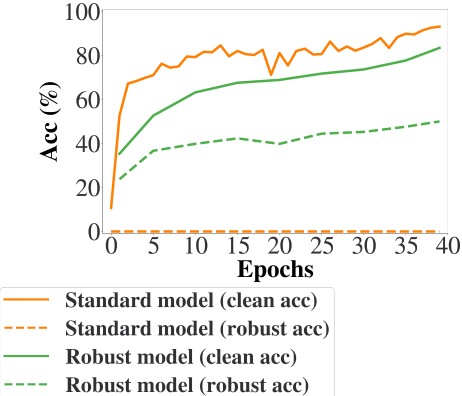

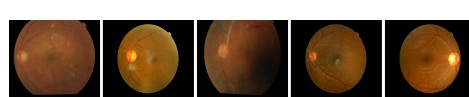

Figure 11: Sample images in the diabetic retinopathy detection training dataset. The key features (e.g. dots and texture) are being detected by the interpretable neurons in Fig 15.

Figure 10: Accuracy vs Epoch for standard and robust model

Probing methodology: We use Broden images as $\mathcal{D}_{probe}$ and for concept set $\mathcal{S}$ we use the broden labels as the concepts can be easily categorized.

**Fine-tuning on medical dataset (section 4.3)**

Setup: We used ResNet-34 backbone pretrained on ImageNet dataset as our feature extractor and used a simple linear layer as the classification head. We trained this network on the diabetic retinopathy classification dataset (APTOS, 2019) (Figure 11) and it achieved an accuracy of 72.77%. We followed the work from (Balaji, 2019) for our experiments. We use Broden as $\mathcal{D}_{probe}$ and broden labels as $\mathcal{S}$.

## B. CONCEPT-MONITOR ALGORITHM

---

**Algorithm 1:** Pseudo code for **Concept-Monitor** for a neuron $n$

---

**Input** : Neuron $n$, concept detector $\phi$, Concept set $\mathcal{S}$, Probing dataset $\mathcal{D}_{probe}$
**Output:** Embedding plot, Concept statistics
**Function** `Concept-Monitor`($\phi$, $\mathcal{S}$, $\mathcal{D}_{probe}$)

    **for** $t$ *from* $1 \rightarrow t_{epoch}$ **do**

        $W_n^t, d_n^t = \phi(\mathcal{S}, \mathcal{D}_{probe}, n)$

        $\lambda_i^n = \text{softmax}(-d_n^t)$

        $u_n^t = \sum_{i=1}^{k} \lambda_i^n f(W_n^t[i]))$

        $\text{plot}(u_n^t)$

        $R_n.append(W_n^t)$

        $D_n.append(d_n^t)$

    **end**

    $stats \leftarrow getStats(R^n, D^n)$

---

## C. VISUALIZING EVOLUTION IN THE EMBEDDING SPACE

Here we use **Concept-Monitor**'s unified embedding space to observe the evolution of few neuron's in layer 4 of ResNet-18 trained on Places 365 dataset as described in Section 3.1. Our embedding space is designed in such a way that it is possible for us to add "anchors" to it, which are positions in the embedding space that represent a particular chosen concept. We show these anchors as red stars in Figure 12. These anchors are fixed through training and mark the region of the embedding space encoding a particular concept so a user may track neuron movements relative to those anchors through training. For brevity we leave out the specific concept labels represented by the anchors in the figure and enumerate them instead. We see that at the beginning most of the neurons

are concentrated around anchors 14,13 and 16 which represent the concepts "grid","dotted" and "porous" respectively, which are low level features. This is expected as the model has just started training and hasn't learnt to encode high level concepts yet. Most neurons move away from this space, except neuron 408 which stays in similar space throughout the training encoding low level textural concepts.

We also would like to highlight the trajectory of neuron 190, which starts from bottom left and slowly moves towards anchor 0 representing the concept minibike. By the end of training, this neuron comes very close to the anchor denoting that it has successfully learnt that concept. This concept of distance to the anchors can also be used as a quick visual aid to tell whether the concepts that the neurons represent are strongly represented or not. If the neuron's concept label is far from the corresponding anchor in the space, we can safely mark that neuron as uninterpretable.

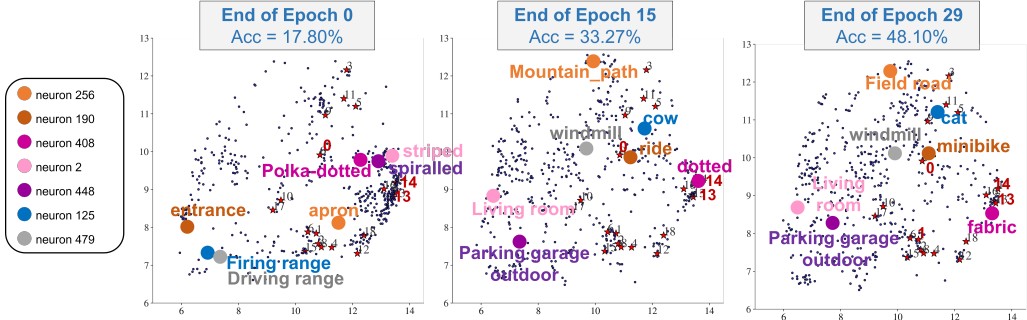

Figure 12: Visualizing evolution of a few neurons using unified embedding space

The labels corresponding to the anchors (red stars) in Figure 12 are 0 - minibike, 1 - exhaust hood, 2 - kitchen island, 3 - leaf, 4 - shower curtain, 5 - net, 6 - pantry, 7 - striped, 8 - countertop, 9 - granite, 10 - forecourt, 11 - cat, 12 - bed, 13 - grid, 14 - dotted, 15 - shower stall, 16 - porous, 17 - aqueduct, 18 - fabric.

### D. CONCEPT MONITOR WITH DIFFERENT PROBING DATASET

As stated in section 3 our method with Clip-Dissect is able to work with any probing and concept dataset. We provide most of our analysis using Broden dataset as it contains a collection of different concept images and hence is able to provide much better results as compared to a limited dataset. Here we provide an example of that by using CIFAR-100 training images as the probing dataset to analyze the same model as section 3. As shown in section 3 and appendix A, neurons 479 represents concept "windmill" and neuron 256 represents the concept "field-road". We now use CIFAR-100 training images to monitor these neurons. From the embedding space in Figure 13 we can see that neuron 256 converges to the "Field" anchor. We also look at the highly activating images for each neuron in Figure 13 and see that for neuron 479 the most activating images are tree like structures across the sky which are the most similar images to windmills in the CIFAR-100 dataset. The point of this exercise is that concept monitor as all other model dissection methods is dependent on the probing dataset, however if we use clip-dissect we are able to use much larger and diverse datasets since we don't require any labelling of images and can simply use entire set of images directly.

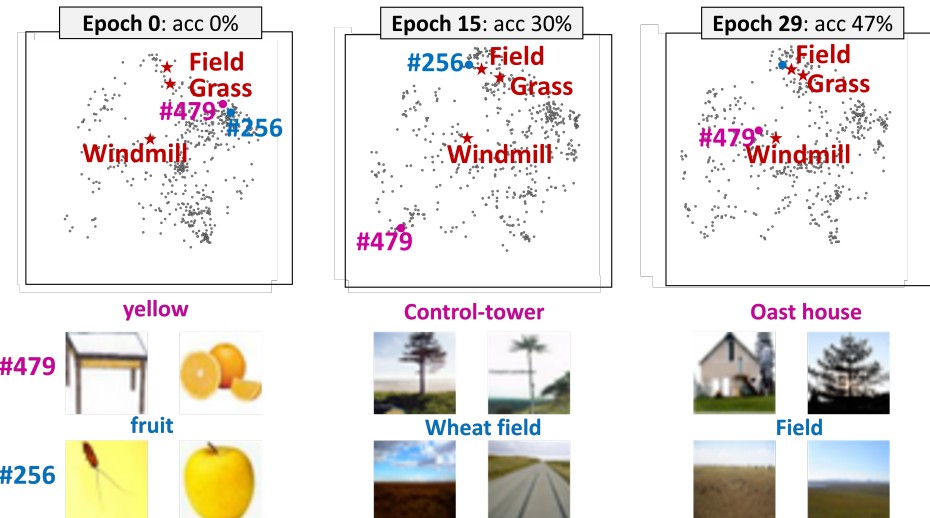

Figure 13: Analysis of Resnet-18 model trained on Places365 dataset using CIFAR100 training images as the $D_{probe}$. The top figure for each epoch is our embedding space where each blue dot represents a neuron in layer 4 of the model. We look at two special neurons, neuron 479 (purple) and neuron 256 (blue). We also project anchor words (shown in red) in the same embedding space. The bottom figure shows the highly activating images for each neuron. (Neuron 479 on top and Neuron 256 at the bottom). We can see neuron 256 gets very close to the anchor 'Field' in the embedding space, as is evident in its highly activating images.

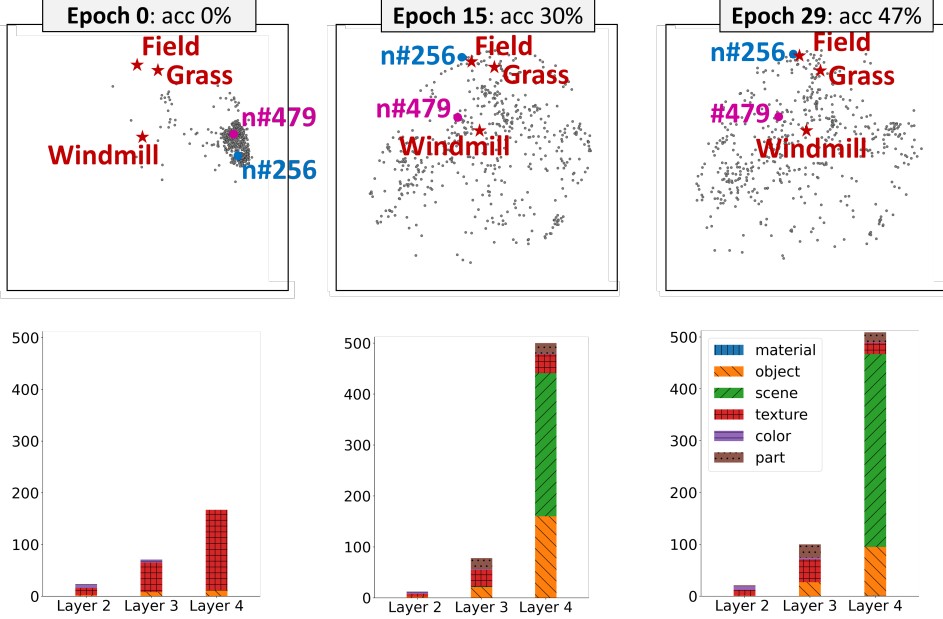

Figure 14: Analysis of Resnet-18 model trained on Places365 dataset using Network Dissection. The top figure for each epoch is our unified embedding space where each gray dot represents a neuron in layer 4 of the model and each red star is an anchor word. Neuron 479 (purple) and neuron 256 (blue) are tracked and are seen to converge to the anchors "Windmill" and "Field" which is consistent with 3. The bottom figure is the number of interpretable neurons in each layer on y axis vs different layers of the model on x axis, divided according to category of their assigned concepts. Here too we see the same trend of increasing higher level concepts in layer 4 and a shift from low level features to high level features in general as training progresses.

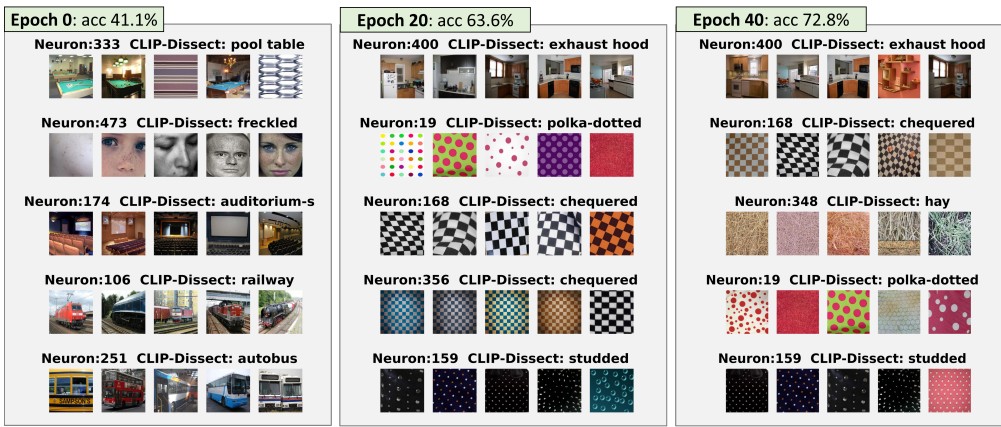

Figure 15: Visualizing concepts learnt by top 5 interpretable neurons of layer 4 of ResNet 34 trained on diabetic retinopathy dataset. We can see that as the training progresses the top concepts shift from complex objects to patterns and textures.

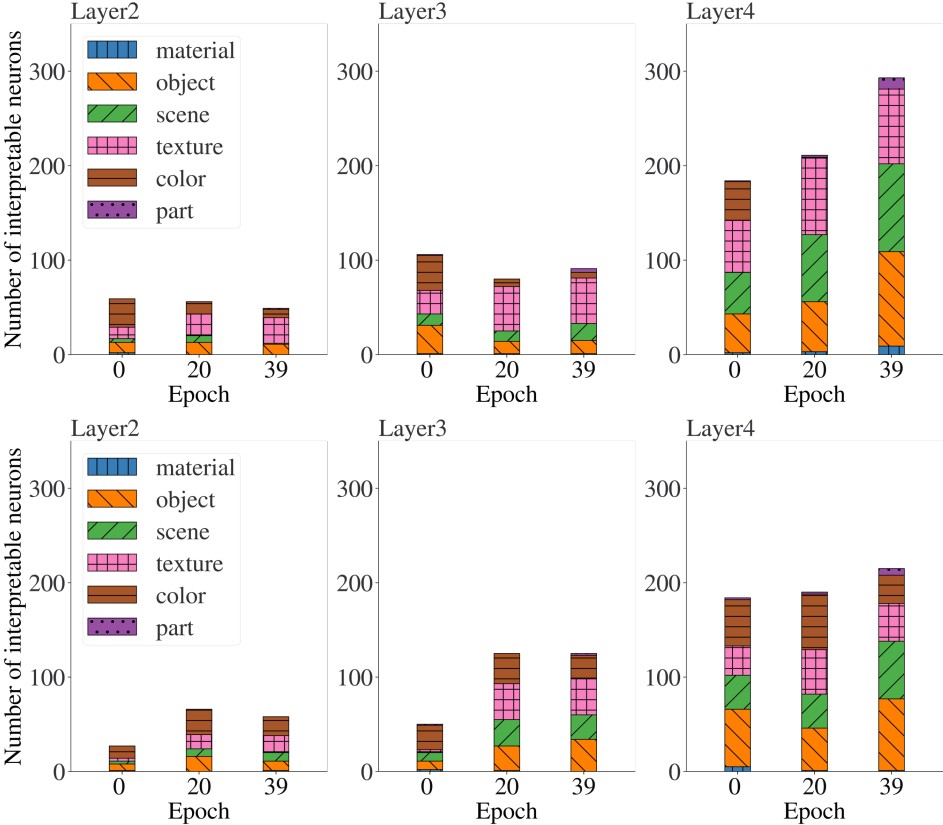

Figure 16: Comparison of the types of concepts learned by standard training compared to PGD-training with $\epsilon$=8/255. We can see the robustly trained model has less interpretable neurons, especially on later layers. Interestingly, there is also a big difference between the types of concepts learned by different models where the PGD trained model still has neurons detecting colors in later layers, while missing concepts detecting textures and materials. Note these figures are the network at the end of each epoch, so epoch 0 is after one epoch of training, not initialization.

