# OpenReview forum: "Demystifying black-box DNN training processes through Concept-Monitor"
_ICLR.cc/2023/Conference — Submitted to ICLR 2023_

### Official Review · Reviewer_d1Hf · 2022-10-16

**Confidence:** 5
**Correctness:** 3
**Technical Novelty And Significance:** 1
**Empirical Novelty And Significance:** 2
**Recommendation:** 3

**Clarity, Quality, Novelty And Reproducibility:**

The contents of the paper are presented in the a clear manner.
The presentation of the content has a good flow.

Regarding novelty, as stated in the summary of the paper, the proposed method is defined by the combination of several existing methods. Beyond the use of an embedding no other novel technical aspect is in place.
Novelty is somewhat limited, at this point the paper seems to be more application oriented.

Reproducibility of the paper is acceptable, implementation details are provided in the supplementary material. In this regard releasing sample code of the implementation of their method with one of the considered concept detectors would have strengthen reproducibility of the proposed method.

**Strength And Weaknesses:**

Strengths
+ Simplicity
+ Clarity
+ Good level of detail

Weaknesses
- Limited technical novelty
- Added value w.r.t. to existing work is not that clear.
- Computational Costs seem to be very high.
- No ablation study
- Limitations are not discussed.

**Summary Of The Paper:**

The paper proposes "Concept-Monitor", a method for analyzing the training process of a given neural network architecture from the perspective of interpretability. More specifically, the proposed method applies concept detectors at every step (i.e. iteration or epoch)  of the training process. The detected concepts are then projected into an embedding space determined by a pretrained large language model. Finally, each neuron is represented by the linear combination of all words/text-labels that have sufficient coverage with it.

The proposed method is validated in four use cases including: 1) Monitoring a regular training process, 2) the analysis of the pruning process in the Lottery Ticket Hypothesis, 3) Adversarial Training, and 4) Fine-tuning.

**Summary Of The Review:**


The manuscript is written in a clear and well organized manner. Its contents are clear and have a good flow. The proposed method is sound, and from a theoretical point of view I can see how it can achieve its goal. Validation of the proposed method is done in a variety of problems/settings.

My main concerns with the manuscript are the following:

The paper claims as third contribution four cases studies where the proposed method is used. I would discourage the claiming this as a contribution since validation of a method proposed in scientific literature is a must. I find good the variety of the analyzed case studies, however, the relevance of their inclusion in the paper must be toned down.

As described in Section 3.1, the proposed method has dependencies on the concept-detector and a large pre-trained language model. On the one hand, this requirements indicate that if the interpretation capabilities of the proposed method would be bounded by the concepts/text modeled by those two components. Moreover, if these two components do not overlap sufficiently with the model whose training process will be analyzed (as when domain-shift occurs), there will be no insight to be obtained.
On the other hand, securing access to sufficiently large components (concept-detectors/language model) would introduce significant computational costs which further explode when you consider the proposed method should be applied in an iterative manner during the training process.
At the moment no limitations of the proposed method are discussed in the paper. Moreover, it would be good to provide some insights regarding the computational cost of the conducted experiments.

Technical novelty of the proposed method is quite limited, as stated earlier, the paper seems to be more application-oriented. However, a consequence of not having an analysis of the computational costs of the proposed method, is that the practical  applicability of the proposed method is not guaranteed.

The proposed method seems to be dependent on a set of parameters: the threshold (\tau) that defines an interpretable neuron, number of selected concepts (k) to represent a neuron. An ablation study of the effect that these parameters have on performance is missing.

Fig.5 shows the number/percentage of interpretable neurons as the IMP process progresses in the LTH case study (Section 4.1). I was wondering how the extraction of this type of insight is exclusive to the proposed method? wouldn't it be possible to obtain similar insights if using one of the concept-detectors considered in the proposed method, in isolation, or any existing DNN interpretation method? A similar argument could be made for the Adversarial Training case study  (Section 4.2)
In this regard a quantitative comparison showing the added value of the proposed method could complement and strengthen the reported results.

For the Adversarial Training case study (Section 4.1) several observations are made regarding the number of interpretable neurons per layer (1) and type of features encoded (2). I was wondering, are these observations made on a single training iteration? or are they common trends observed along different runs?

Finally, while the case studies considered in the evaluation are interesting, the paper seem to deviate from the analysis of the proposed method and dedicate too much space to present/discuss points exclusive to those case studies.

[Minor] Plots in Figure 3 are missing axis labels.

---

### Official Review · Reviewer_Hunn · 2022-10-24

**Confidence:** 5
**Correctness:** 1
**Technical Novelty And Significance:** 1
**Empirical Novelty And Significance:** 1
**Recommendation:** 1

**Clarity, Quality, Novelty And Reproducibility:**

- Clear presentation
- Novelty is too limited
- Results are straightforward

**Strength And Weaknesses:**

+ Interesting idea of visualizing neurons on a temporal / training scale
But
- no real use for such tool
- unclear motivation
- already known from the community

I would recommend to use the tool to drive deeper research directions e.g., how this could guide / better shape re-training on the go by having systems that could be recommended particular images - combining the output with GAN architecture would be interesting.

**Summary Of The Paper:**

This paper presents a general framework called Concept-Monitor to uncover the black-box DNN training processes. The paper is built on top of existing work capturing concept based representation, but rather on a temporal / training scale, which sounds interesting but of limited use.

**Summary Of The Review:**

An interesting paper but coming too late as most of the findings are already known. This could be interesting if the tool was contextualized in a deeper research question.

---

### Official Review · Reviewer_dpnA · 2022-10-31

**Confidence:** 3
**Clarity, Quality, Novelty And Reproducibility:** The paper is well-organized and easy …
**Correctness:** 3
**Technical Novelty And Significance:** 3
**Empirical Novelty And Significance:** 3
**Recommendation:** 5

**Strength And Weaknesses:**

Strength:
1.	Unlike the previous works on neural network explanations that can interpret a static neural network, the proposed Concept-Monitor can produce human-interpretable visualization during training and help us better understand the training process of black-box neural networks.
2.	The proposed method is simple and easy to reproduce. It is training free and easy to be adapted to new model architectures.
3.	The new findings with Concept-Monitor on adversarial training and network pruning are interesting and provide a new perspective to understand other techniques in deep network training.

Weakness:
1.	The proposed method is a simple combination of network dissection and CLIP-dissect to define the interpretable neuron. I would like to see more clarification to differentiate the proposed method and those two classic methods. Please explain more on the technical contributions and motivations compare with network-dissection and CLIP-dissect.
2.	The proposed method should involve intensive computation costs to get the concept of some interpretable neurons and it may limit the practical usage of the proposed method. I would like to see the computation analysis and potential improvement.


**Summary Of The Paper:**

The authors propose a novel method to interpret the black-box neural network training process. Extensive experiments demonstrate the proposed Concept-Monitor can help find some intriguing properties of adversarial training and network pruning.

**Summary Of The Review:**

See more details in the strength and weaknesses.

---

### Official Review · Reviewer_UEGP · 2022-11-04

**Confidence:** 3
**Correctness:** 3
**Technical Novelty And Significance:** 2
**Empirical Novelty And Significance:** 3
**Recommendation:** 5

**Clarity, Quality, Novelty And Reproducibility:**

This work is marginally novel, however, regarding clarity, there are some details related to concept sets, concept detectors, and the language model encoder which are missing.

**Strength And Weaknesses:**

Strengths:

The paper is overall clear and the topic is well-motivated.

Weaknesses

Although the approach is simple and lightweight, the paper fails to present a systematic and rigid approach to support their findings and observations in the three case studies. In particular, while the authors compared their method to [Park et al., 2022] in terms of efficiency, they did not provide any comparisons with this approach to show the consistency of their method. It is also necessary to show if the results are consistent across different initializations, different models, datasets, etc.

Below I listed some of the problems in the paper:
- It requires a reference. "a good progression of concepts learnt might indicate a well trained model."
- The paper claims: "they can pause training or modify hyper-parameters when they see neurons grouping up or not spreading out in the semantic space." It would be great to show if it would practically work for poorly trained models. Does a poorly-trained model show odd concepts in terms of neuron interpretability?
- The authors need to provide more details on training the encoder for robustness of the findings to the chosen thresholds \tau.
- The results of Figure 4 and Figure 14 are not consistent. For example, for Epoch 0 the patterns are entirely different. In addition, Layer 3 of epoch 29 is very different as "part" is not detected in Figure 4.  The number of interpretable neurons for Figure in layer 2 is notably higher than in Figure 14.
- There are some questions that can be further discussed in case studies: For example in adversarial training, what happens if a model is more robust than the other model? Or how do the results shift for different types of adversarial training such as PGD with different number of iterations or randomized smoothing.

**Summary Of The Paper:**

The paper introduces a mechanism to describe and quantify the interpretability of neurons in the training process and the experiments are mainly devoted to demonstrating how the concepts shift as the training progresses. For this purpose, the authors utilize the description of neuron representations in the literature to calculate the embedding weights after encoding the top k concept words for neurons.

**Summary Of The Review:**

I believe the approach is novel and efficient but the generalizations of findings and observations in the case studies are limited due to the
lack of enough comparisons and ablation cases.

---

### Official Review · Reviewer_kK6R · 2022-11-09

**Confidence:** 5
**Correctness:** 2
**Technical Novelty And Significance:** 1
**Empirical Novelty And Significance:** 2
**Recommendation:** 1

**Clarity, Quality, Novelty And Reproducibility:**

The paper is clearly written, but the quality and novelty of the contribution are extremely limited. The scientific insights are not likely to reproduce across settings and tasks, therefore not offering any useful information without re-experimentation in the domain/dataset/architecture of interest. In addition, without open-sourcing or API for the Concept Monitor product, one would have to completely reimplement the system, without being able to take advantage of the work of the authors.

**Strength And Weaknesses:**

Strengths:
- Neat packaging of prior research contributions into a useful tool for interpretability
- Interesting selection of use-cases (e.g. lottery tickets, adversarial training)
- Clarity of exposition

Weaknesses:
- Concept Monitor is not, itself, a novel interpretability method and does not provide any novel insight into the inner workings of a network. It consists, instead, of a system built on top of various interpretability methods to better track and visualize the output of these other methods over time during training.
- It is only applied, in this paper, to the field of image processing, and, specifically, to a very small set of architectures and datasets, with no replication of results across seeds nor ablations across tasks and setups, so there is no evidence of generality of the observations presented in this work.
- It requires a predefined set of concepts, which might be limiting. Since it's only applied to image processing, accepted concept sets do exist and capture high level clusters of concepts focusing on texture, material, color, etc. However, in other context, the weakness of this approach has already been pointed out as undesirable.
- The paper limits itself to interpreting models at the neuron level, which has been criticized in the literature as not being the right level of abstraction for interpreting neural networks. It is possible that Concept Monitor would work at other levels of abstraction as well (layer, circuit, branch, etc.) because it ultimately delegates the extraction and assignment of interpretable concepts to interpretable units to other methods in the literature, therefore only building on top of the outputs of these methods. However, no discussion nor demonstration of this is shown in this work.
- Poor coverage of the related literature. Among many others, this paper is missing references and discussion of the work on circuits (e.g. Olah et al, 2020), neuron specialization (Goh et al., 2021, Nguyen et al., 2016, Cammarata et al. 2020), polysemanticity and superposition (Elhage et al., 2022), neuroscience, and much more. The work is also lacking a discussion of bias in word embeddings, which it heavily relies on.
- The insights are based on one-time observations and demos, with no statistical significance associated with the results. This is because experimental evidence is intended to be used as proof of concept for the utility of the Concept Monitor service, and not to extract generalizable insights of scientific value. Most of the discussion in the paper, however, focuses on the insights extracted from these one-time experiments, attempting to draw conclusions about, say, a particular network rewound to epochs 0, 5, or 16, but utilizing results such as the ones in Figure 5 with no error bars or confidence bands.
- Insights about the changes in relative importance of texture vs higher level concepts in adversarially trained models are already well known.

**Summary Of The Paper:**

The paper presents a visualization service called Concept Monitor that can be used to inspect the evolution of concepts associated with single neurons throughout the training of a vision model. The system's interoperability allows for it to be utilized with different concept detection algorithms from the literature, such as Network Dissection (Bau et al., 2017) and CLIP-Dissect (Oikarinen and Weng, 2022), among others. The authors use the embedding space of a pretrained language model as their unified embedding space onto which concept words are projected; this has the advantage of providing stationary anchor points that don't vary over time, as this embedding function is pretrained and fixed -- only neurons will move across this space over time due to training. The solution offered in this contribution intends to shed light onto training dynamics by providing a more user-friendly visualization tool.

**Summary Of The Review:**

The paper is of insufficient quality for publication. Major factors contributing to this opinion include: the lack of novelty, the lack of empirical significance to back up the insights extracted through the experiments presented in this paper, and the inadequate contextualization and representation of the existing literature in this domain.

---

### Decision · Program_Chairs · 2023-01-20

**Decision:**

Reject

**Justification For Why Not Higher Score:**

The reviewers all noted weak experimental evaluation, insufficient citation of the previous literature, and limited novelty.

**Justification For Why Not Lower Score:**

N/A

**Metareview: Summary, Strengths And Weaknesses:**

The author introduce the concept monitor, a method that tracks the evolution of concepts of single neurons throughout training process. Reviewers noted the simplicity of the method and clarity of exposition, but also found serious weaknesses. These include limited novelty, limited experimental evaluation to better understand the performance of the method, and insufficient citation of previous literature.